# IoT and ML-based Water Flow Estimation using Pressure Sensor

Maulesh Gandhi, Sannidhya Gupta

**Abstract**

This study presents an Internet of Things (IoT)-based system that utilises machine learning (ML) techniques to estimate water flow through pipes based on pressure. The system incorporates an ESP-32 microcontroller, a Danfoss MBS 3000 pressure sensor, and a flow meter deployed at three locations to collect data for three months. To model the relationship between pressure and flow rate, ML algorithms such as linear regression (LR), support vector regression (SVR), and convolutional neural network (CNN) were trained, analysed, and compared. By establishing a model to estimate the flow rate based on pressure, the need for a flow meter in the setup can be eliminated. The system's low-cost, easy-to-implement, and non-invasive nature makes it suitable for widespread adoption in residential areas, offering a promising solution for optimising water distribution and reducing water wastage.

## 1   Introduction

The scarcity of good quality water for drinking and residential purposes is among the most significant threats in recent years. Most water wastage occurs because of a lack of water monitoring. Water, being precious, needs to be preserved and appropriately utilised. For this, accurate and real-time monitoring is essential. By monitoring water levels, quality, and ecosystem health, we can make informed decisions to safeguard public health, protect the environment, and promote sustainable practices.

With the advancements in the Internet of Things (IoT) [1], many digital counterparts have come up in recent years to automate the meter reading mechanism by utilising cloud servers. For water metering, IoT is a game-changer, offering real-time data collection from sensors placed strategically across water sources and distribution systems. This interconnected network enables remote monitoring, rapid leak detection, and early identification of contamination events with minimal human intervention needed. IoT-driven water monitoring paves the way for a more efficient and resilient water management system. Moreover, IoT-powered water monitoring can integrate with advanced analytics and machine learning algorithms to derive meaningful patterns and trends from the data. This empowers stakeholders to make informed decisions, develop proactive strategies for water management, and implement sustainable practices.

Some research has been conducted on IoT-based water monitoring [2, 3, 4, 5, 6, 7]. Most of the research has been focussed on leakage detection and prevention of water wastage. [2] explores existing technologies for detecting leakages using vibration, acoustic, flow and other types of sensors. [3] demonstrates a wireless sensor network for identifying the leak's location. Identifying the exact place of leakage using an acoustic sensor was done in [5]. Literature [4] and [6] propose IoT-based pressure monitoring systems and leak detection techniques based on the pressure variations at different locations. [7] proposes retrofitting the existing analog meters with a low-cost solution by utilising ML algorithms and image processing, which is more feasible and affordable to the customers. Still, the dependency on the existing meter and its dial orientation is a challenge.

The work in this project is different from the work in literature [2, 3, 4, 5, 6, 7] in that the water flow rate through a pipe is estimated using a pressure sensor and machine learning (ML). This way, the pressure sensor can replace the flow meter in most situations. There are several advantages of using a pressure sensor compared to a water flow sensor. For example, pressure sensors can be non-invasive as they often measure pressure without direct contact with the fluid. Pressure sensors are generally more cost-effective than water flow sensors. They often have a simpler design and require less specialised equipment for installation. The particular contributions of this project are

- An IoT and ML-based mechanism is proposed to estimate water flow using a pressure sensor.

- For this, an IoT node measuring water pressure and flow is developed in IIIT-H to collect these parameters at the same location. The node is compared against an off-the-shelf pressure indicator (or meter).

- Three such nodes are deployed in different locations on the IIIT-H campus, Hyderabad, India, collecting data over three months.

- Machine learning algorithms such as linear regression (LR), support vector regression (SVR), and convolutional neural network (CNN) [8] are employed to train models to estimate the flow rate from pressure values.

- The trained models are tested through k-fold validation [9] using evaluation parameters of root mean square error (RMSE) and accuracy.

## 2   Goals

The primary goal of this project is to develop an innovative IoT-based system that leverages machine-learning techniques to estimate water flow through pipes based on pressure. This goal is driven by the urgent need to address the significant challenges associated with water scarcity, wastage, and the lack of low-cost, real-time water monitoring solutions in residential areas. To achieve this overarching goal, the following specific objectives have been established:

1. Cost-Effective and Non-Invasive Solution - Design a solution that is cost-effective, easy to implement, and non-invasive, ensuring it can be readily adopted in residential areas without significant infrastructure changes to immediately facilitate real-time monitoring.

2. Long-Term Data Collection - Collect pressure and flow data from three strategically deployed IoT nodes over a period of three months to ensure the reliability and robustness of the solution's performance.

3. Pressure-Based Flow Estimation - Using the collected data, create a mechanism to estimate water flow rates through pipes solely based on pressure data, eliminating the need for traditional flow meters in most scenarios.

4. Machine Learning Model Development - Utilize machine learning algorithms, including linear regression (LR), support vector regression (SVR), and convolutional neural networks (CNN), to train models that establish the relationship between pressure and flow rate.

5. Comparative Analysis - Analyze and compare the performance of the machine learning models in estimating flow rates, using evaluation parameters such as root mean square error (RMSE) and accuracy through k-fold validation.

The fulfilment of these objectives is expected to result in a low-cost, efficient, and easy-to-adopt solution that offers promising potential for optimizing water distribution, reducing water wastage, and promoting responsible water usage in residential areas.

## 3   System Architecture and Design

### 3.1   Hardware

The block architecture of the node developed is shown in Fig. 1. Every node consists of a Danfoss MBS 3000 pressure sensor[10], a Shenitech ultrasonic flow meter[11], and an ESP32[12] for measuring the pressure and flow rate of water going through pipelines. The pressure sensor gives a current output of 4-20 mA based

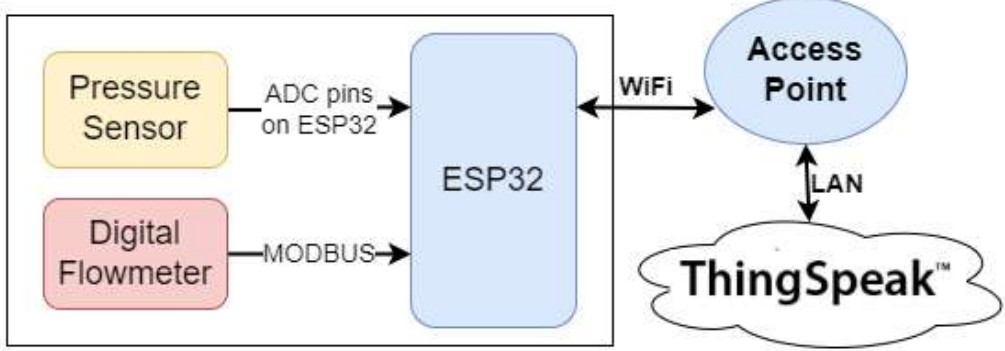

Figure 1: Block architecture of the node

Table 1: Specifications of sensors used in the developed node

| Sensor | Parameter | Resolution | Relative error |
|---|---|---|---|
| Danfoss MBS 3000 [10] | Pressure | 0.000001 bar | 0.5% FS (typ.) 1% FS (max.) |
| 280W-CI ultrasonic water meter [11] | Total flow, flowrate | 0.01 kL 0.001 kL/h | ± 5% FS (max.) ± 5% FS (max.) |
| AI-PRI digital pressure indicator [14] | Pressure | 0.01 bar | 0.25% FS |

on the pressure inside the pipes, which is converted to a voltage level of 0.6-3 V using a 150$\Omega$ resistor, and then the pressure voltage is read by the ESP32. The flow meter and pressure indicator readings are read using MODBUS communication. An external adaptor provides the 12V required for the pressure sensor, and a regulator converts it to the 5V power supply required by the ESP32. The software code in the ESP-32 updates the values to the Thingspeak[13] cloud server using Wi-Fi. An additional off-the-shelf AI-PRI digital pressure indicator by Ace instruments[14] is deployed at one of the deployment locations. This pressure indicator aims to estimate the pressure values more accurately and validate our pressure readings.

Every node consists of a Danfoss MBS 3000 pressure sensor [10], a Shenitech ultrasonic flow meter [11], and an ESP32 [12] for measuring the pressure and flow rate of water going through pipelines. The pressure sensor gives a current output of 4-20 mA based on the pressure inside the pipes, which is converted to a voltage level of 0.6-3 V using a 150 Ω resistor, and then the pressure voltage is read by the ESP32. The flow meter and pressure indicator readings are read using MODBUS communication [?]. An external adaptor provides the 12V required for the pressure sensor, and a regulator converts it to the 5V power supply required by the ESP32. The software code in the ESP-32 updates the values to the Thingspeak [13] cloud server using Wi-Fi. An additional off-the-shelf AI-PRI digital pressure indicator by Ace instruments [14] is deployed at one of the deployment locations. This pressure indicator aims to estimate the pressure values more accurately and validate our pressure readings.

## 3.2   Field deployments

With the help of the above-described hardware setup, three nodes were deployed at different locations across the IIIT-H campus: one at the pump house pumping water for domestic usage to most of the college hostels (PH0100) and two at separate outputs of another pump house pumping drinking water to two different faculty quarters (PH0302 and PH0303). The locations of deployed nodes are shown in Fig. 2, and Fig. 3 shows images of the PH0302 node.

## 3.3   Software

### A. Data description

Figure 2: Locations of deployed nodes

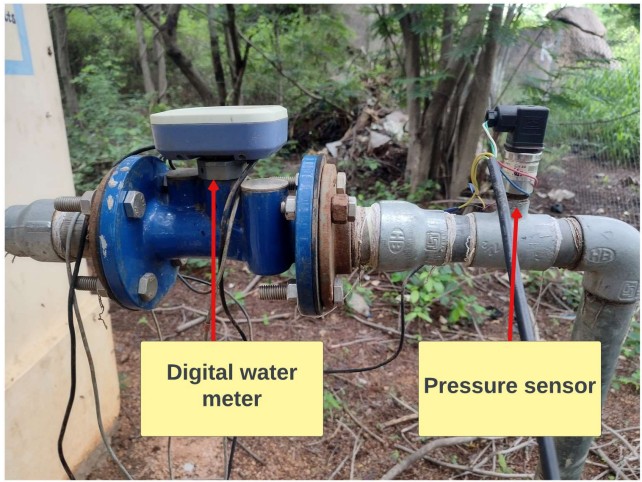

(a) Pressure sensor and digital watermeter

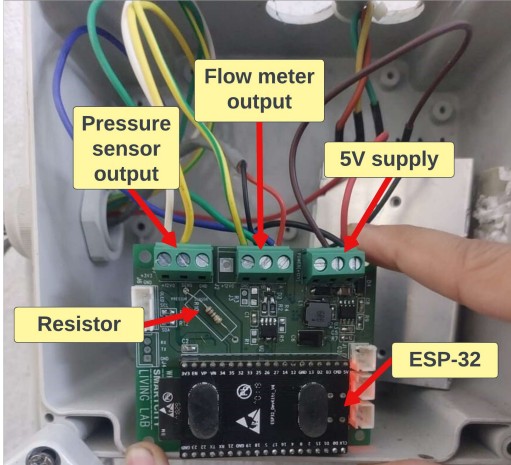

(b) IP65 box containing microcontroller and SMPS.

Figure 3: One of the deployed units. The digital meter and pressure node are connected to the microcontroller in an IP65 box inside the pump house.

Data was collected over three months for the three nodes at an interval of 90 seconds. Over 75,000 data points were collected for each node. Each data point consists of a timestamp, pressure voltage, instantaneous flow rate, and total water flow from the time of installation. Only the PH0100 node has extra pressure readings from the AI-PRI digital pressure indicator, which converts the pressure sensor readings directly to pressure values with an accuracy of 0.25 % of Full scale (FS) and gives it to the microcontroller. The indicator has been calibrated using an advanced pressure factor marker to estimate pressure more accurately and advantageously. Since these values are more accurate than the measured pressure values, which have an accuracy of 0.5 % of FS, they are used as ground truth pressure values to validate the measured values. From the collected data, we calculated two more parameters for further processing:

1. Calculation of pressure from pressure sensor output: The pressure sensor gives an output current of 4-20 mA linearly corresponding to an input pressure of 0-16 bar [10]. The ADC pins of the ESP-32 take input voltages ranging from 0-3.3V and convert them to integer values ranging from 0 to 4095. A 150 Ω resistor was used to convert the current to a 0.6-3 V voltage to allow high resolution of the readings. An ADC pin on ESP-32 reads the voltage value and gives a corresponding integer value. The pressure P (in bars) is calculated from the obtained ADC output A (in V) using the given equation:

$$P = \frac{A \cdot 3.3 \cdot 1000}{4095 \cdot 150} - 4$$

2. Calculation of flow rate from total flow readings: The flow rate readings obtained from the meter may contain outliers due to their instantaneous nature. Hence, the total flow readings and corresponding timestamps were used to calculate the flow rate at each data point to get more accurate flow rate values. They were calculated as the difference between two consecutive total flow readings (in kL) divided by the difference in their corresponding timestamps (in hours).

**B. Data cleaning and preprocessing**

The dataset collected from the deployed nodes underwent several preprocessing steps to enhance the accuracy and reliability of the models. The preprocessing steps included:

1. Outlier removal: Outliers were defined as data points deviating significantly from their respective clusters' median. k-means clustering was used to identify outliers based on the Silhouette coefficient scores. The Silhouette coefficient evaluates the quality of clusters created using clustering algorithms such as k-means. Since higher values indicate more well-separated clusters, the optimal number of clusters was chosen to maximise the coefficient (k was chosen as two). The detected outliers were replaced with the median values of their clusters.

2. Data smoothing: A moving average technique with a window of size five was applied to smooth the data and reduce noise. The smoothing helped remove short-term fluctuations and highlight the dataset's underlying trends.

These preprocessing steps were crucial in improving the dataset's quality by reducing noise, removing anomalous data points, and capturing the underlying patterns more accurately. This preprocessed data is used for further processing.

# 4   Tools used for data analysis

## 4.1   Correlation coefficient

Pearson's correlation coefficient was calculated between the calculated pressure and flow rate values for all nodes and the calculated pressure and the ground truth pressure values for the PH0100 node. Pearson's correlation coefficient measures the strength of a linear association between two variables. A coefficient close to 1 or -1 suggests a strong positive or negative linear relationship, respectively, while a coefficient close to 0 indicates no significant linear relationship. Pearson's correlation coefficient detects linear relationships between variables but may not be as effective in detecting non-linear relationships.

## 4.2 Training Machine Learning models

Three different ML algorithms were employed for training LR, SVR, and CNN, which are described in brief below:

**1) Linear regression (LR):**

LR is a statistical method used to model the relationship between a dependent variable and one or more independent variables by fitting a linear equation to the observed data. It served as a baseline model for comparison with more complex algorithms. LR model aimed to capture the linear relationship between the pressure readings and the corresponding flow rates.

**2) Support vector regression (SVR):**

SVR is a robust regression algorithm that captures complex relationships between input features and the target variable. It leverages support vector machines to approximate the mapping function. SVR can effectively model non-linear data patterns, which this study employed to capture non-linear relationships between pressure sensor readings and flow rates.

**3) Convolutional neural network (CNN):**

CNNs are deep learning models widely used for image and sequence analysis tasks. This study utilised a CNN model to leverage the complex patterns in the dataset. Given the high correlation between pressure values and flow rate, our study employed a basic CNN architecture consisting of two convolutional layers, a Rectified Linear Unit (ReLU) activation, and one final dense layer. The final dense layer consists of a single neuron since the model performs a regression task (predicting a single continuous value). The CNN models were trained using the Adam optimiser at the default learning rate of 0.001 for 100 epochs using the data from each node and evaluated using the testing set.

The models for each node were trained on data collected at that node. The dataset included pressure sensor readings and corresponding flow rates measured by the water flow meter. The training set was used to train each model, and the testing set was used to evaluate their performance. The models were assessed based on various evaluation metrics, which provided insights into the estimative capabilities of the trained models.

By employing LR, SVR, and CNN models, the study aimed to capture the relationships between pressure sensor readings and flow rates from different perspectives, allowing for more accurate and robust estimations.

# 5 Performance Evaluation and Testing Results

## 5.1 Testing

The performance of our trained ML models was evaluated using k-fold validation. It is a technique commonly used to assess the generalisation ability of a model by dividing the dataset into k equally-sized folds or subsets. A model is trained on k - 1 folds, and its performance is evaluated on the remaining fold. This process is repeated k times using a different fold as the test set, and the evaluation metrics are averaged across all iterations. Some benefits of k-fold validation are reduced bias, better parameter tuning, and improved generalisation. Using k-fold validation, more accurate and reliable estimates of the model's performance were obtained.

## 5.2 Evaluation metrics

Our machine learning models' performances were evaluated using root mean squared error (RMSE) and accuracy/coefficient of determination ($R^2$). The RMSE measures the root average squared difference between the estimated and actual flow rate values (unit is kL per hour). In contrast, $R^2$ measures the proportion of variance in the dependent variable that is estimable from the independent variable. The $R^2$ is defined as $(1 - \frac{u}{v})$, where $u$ is the residual sum of squares $\sum(y_{\text{true}} - y_{\text{pred}})^2$ and $v$ is the total sum of squares $\sum(y_{\text{true}} - \bar{y}_{\text{true}})^2$.

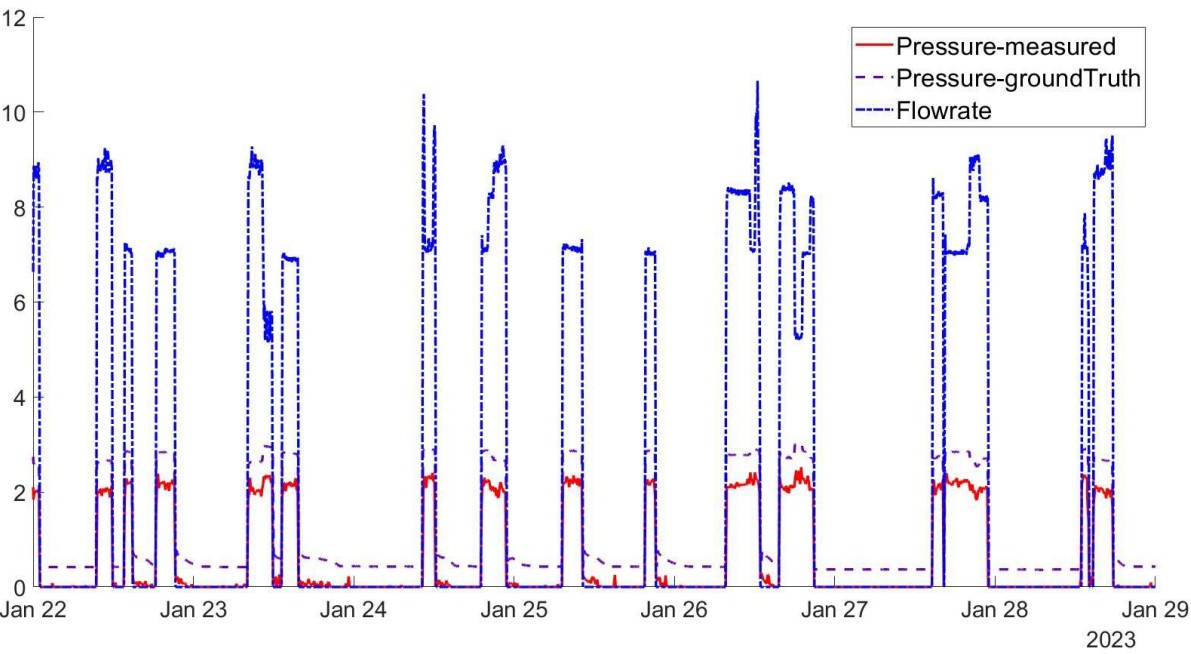

Figure 4: Processed Data

Table 2: Correlation coefficients between pressure and flow for different locations

|  | PH0100 | PH0302 | PH0303 |
|---|---|---|---|
| **Pressure (measured) and flowrate** | 0.970 | 0.973 | 0.990 |
| **Pressure (measured) and pressure (ground truth)** | 0.993 | NA | NA |

## 5.3  Results

**A. Time Series Data:**

Fig. 4 shows the time-series plot over a week, obtained after applying our preprocessing steps to the original data. Fig. 5 shows the same data after normalising the processed data for better visualisation by scaling it to a range of 0-1. Both the plots show a high correlation between pressure (ground truth), pressure (measured), and flow rate, which can also be observed from the correlation coefficients of all nodes, given in Table 2. The high correlation observed is the basis for the proposed algorithm. Also, due to the high correlation observed between the measured pressure values and the ground truth pressure values obtained from the pressure indicator, it can be concurred that there is no need for data calibration.

## 5.4  Time Series Data

**B. Machine Learning Results**

After calculating the correlation coefficient between our variables, we applied LR, SVR, and CNN models to model the relationship between the measured pressure and the estimated flow rate. The results obtained after training the models for all three nodes- PH0100, PH0302, and PH0303 are presented in subplots 6a, 6b, and 6c respectively. By analysing the scatterplots of training and testing data, we observed non-linear patterns, particularly distinguishable in the scatterplot of node PH0100. This non-linearity could be attributed to several factors, such as the inherent behaviour of the water flow system, variations in pipe conditions, or the influence of other variables not accounted for in the analysis. These factors might introduce complexities

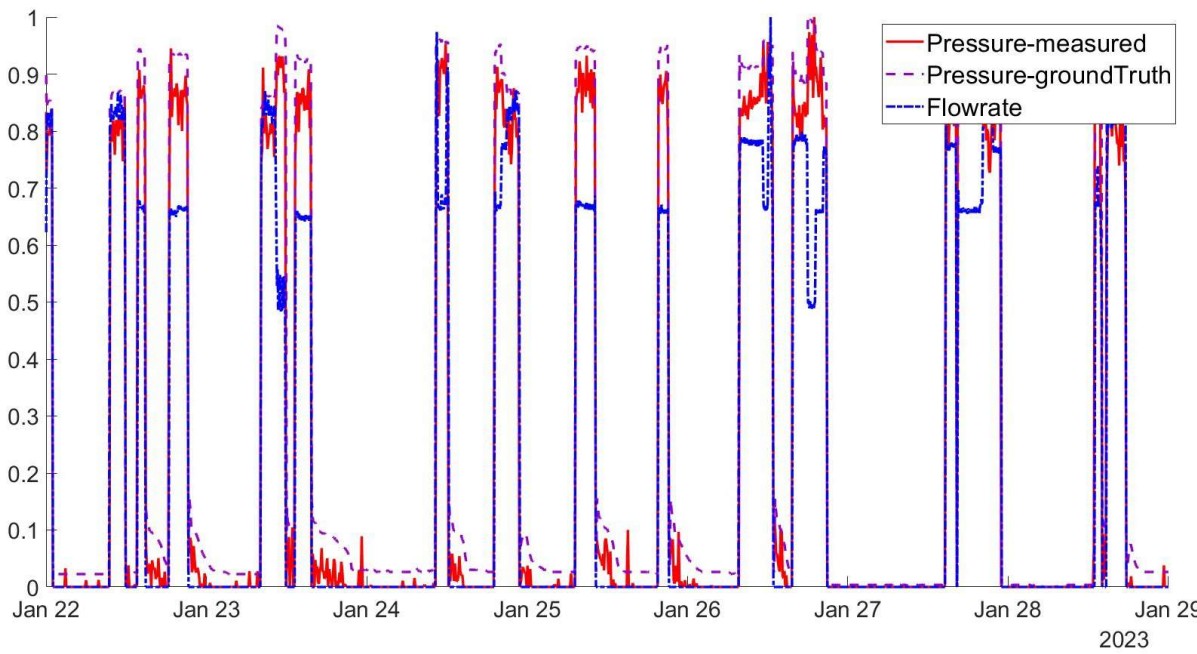

Figure 5: Normalized Data

Table 3: Accuracies of ML models trained

| Model | PH0100 | PH0302 | PH0303 |
|:-----:|:------:|:------:|:------:|
| LR | 0.942 | 0.946 | 0.980 |
| SVR | 0.969 | 0.981 | 0.984 |
| CNN | 0.974 | 0.984 | 0.984 |

that a simple linear relationship between pressure and flow rate cannot adequately capture. The accuracies and RMSE errors obtained after applying k-fold validation with k = 10 for the ML models are shown in Tables 3 and 4, respectively. These results provide a comprehensive overview of the performance of each model in estimating flow rates based on pressure readings. LR achieved reasonable accuracies above 90% for all nodes, indicating a decent fit to the linear relationship. However, its performance may be limited in capturing the non-linear variations observed in the scatterplots, especially for PH0100. In contrast, SVR and CNN models outperformed LR, offering similar higher accuracies and lower RMSE values. These models' ability to handle non-linear relationships allowed them to approximate the underlying complexities of the pressure-flow rate correlation, as evidenced by the improved accuracy on the PH0100 dataset with pronounced non-linear patterns. Overall, the results suggest that both SVR and CNN models are better suited for water flow estimation tasks, especially when dealing with non-linear relationships between pressure and flow rate. These models offer enhanced accuracy and robustness, enabling more reliable water management strategies, even in complex and non-linear pressure-flow dynamics.

Table 4: RMSE (in kL) values of ML models trained

| Model | PH0100 | PH0302 | PH0303 |
|:-----:|:------:|:------:|:------:|
| LR | 0.873 | 0.450 | 0.302 |
| SVR | 0.635 | 0.253 | 0.269 |
| CNN | 0.579 | 0.232 | 0.273 |

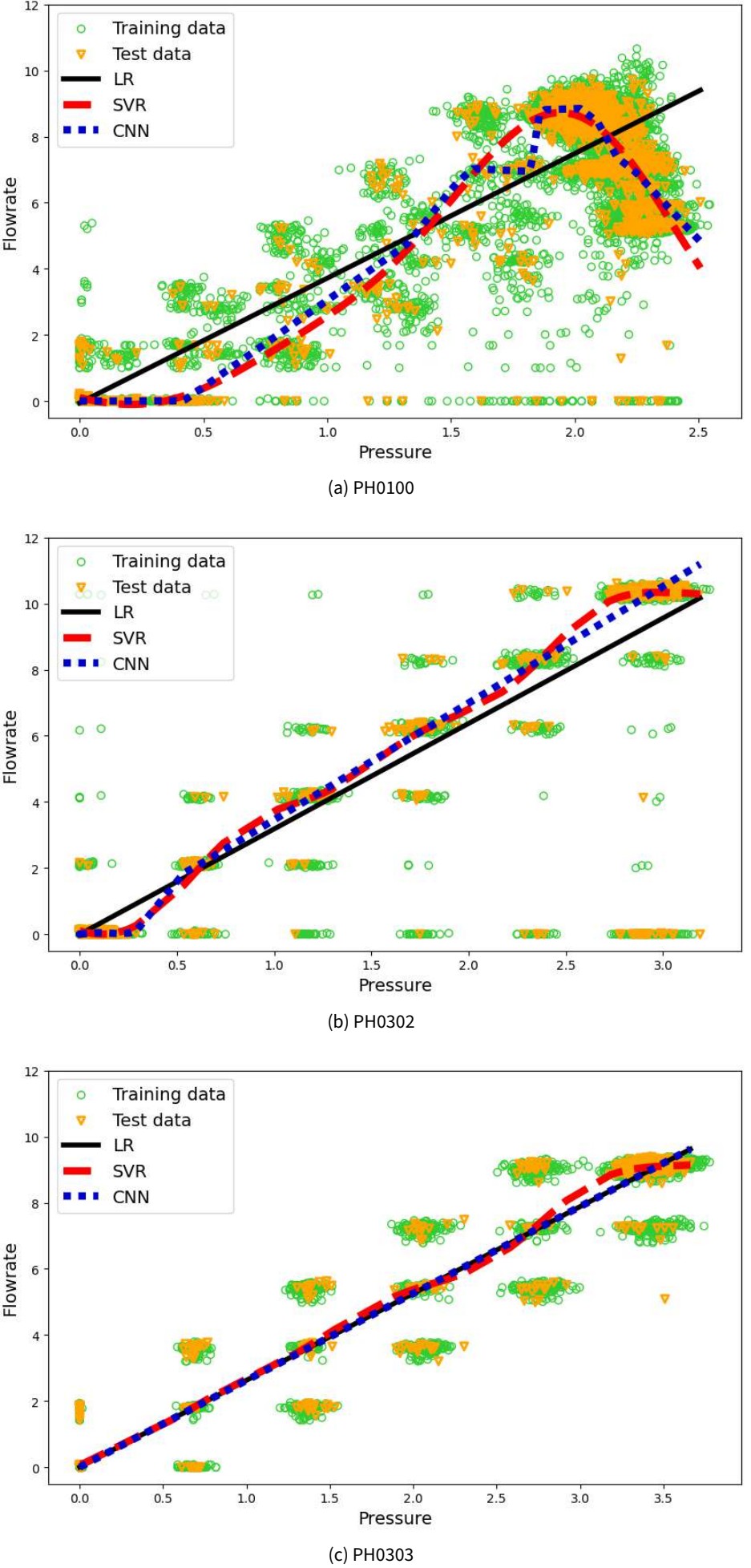

(a) PH0100

(b) PH0302

(c) PH0303

Figure 6: Results of ML models trained

# 6   Concluding Remarks and Avenues for Future Work

In conclusion, this study's findings demonstrate the effectiveness and reliability of the proposed IoT system in accurately estimating water flow based on pressure readings, which were performed on data collected from actual deployments. By establishing an ML model that accurately models the correlation between pressure and flow rate, reliance on a flow meter to provide flow rate readings can be eliminated, simplifying the water pipeline system and reducing costs and maintenance requirements. Models based on LR, SVR, and CNN were trained, and their performances were evaluated and compared. All the models gave accuracies above 90%, with CNN models giving the best results. This IoT-based water flow estimation system offers a promising low-cost solution for monitoring water distribution and minimising water wastage. Future works can include increasing the accuracy of the models and performing analysis over a longer time period.

# 7   Availability

URL for source codes: https://github.com/MauleshGandhi/Pressure_analysis

URL for video: https://iiitaphyd-my.sharepoint.com/:v:/g/personal/maulesh_gandhi_research_iiit_ac_in/EbrhM9WZPwdMhKGP8ArMOu4BjdO4zQgDGU1JonA3OkgH5A?e=Yv5x7Z

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
