# OpenReview forum: "IoT and ML-based Water Flow Estimation using Pressure Sensor"
_helsinki.fi/ESPC/2023/Competition — ESPC 2023 LongPresentation_

### Official Review · Reviewer_xCZ8 · 2023-11-16

**Rating:** 3
**Confidence:** 2

**Summary:**

The article presents an IoT system that estimates water flow through pipes based on pressure sensors with the use of machine learning. The system was fully designed, implemented, and deployed for data collection for three months.

**Strengths:**

- Actual design, implementation, and deployment of the system (3 nodes)
- Continuous data collection (for 3 months)
- Various ML algorithms are used and compared
- Training and evaluation procedures are explained in detail
- Report is generally well written

**Weaknesses:**

- It would be nice to see a better analysis of model performance, e.g. including residual plots
- Section 3.1 contains some repetition

---

### Official Review · Reviewer_bbrL · 2023-11-18

**Rating:** 3
**Confidence:** 3

**Summary:**

This project introduces an IoT-based approach that uses ML methods to estimate water flow through pipes based on pressure. The project uses an ESP-32 microcontroller, a Danfoss MBS 3000 pressure sensor, and a flow meter deployed at three locations to collect data for the period of three months. To model the relationship between pressure and flow rate, the project utilizes three ML algorithms including linear regression (LR), support vector regression (SVR), and convolutional neural network (CNN).

**Strengths:**

The idea of employing a software/ML-based approach that replaces physical tools. Implementing the proposed ML-based approach that estimates the water flow rate based on pressure, there won’t be a need for a physical flow meter.

**Weaknesses:**

The project lacks a rich explanation about the architecture, sensor systems, and data collection.

---

### Official Review · Reviewer_gwv9 · 2023-11-18

**Rating:** 3
**Confidence:** 3

**Summary:**

In this work, authors have developed a prototype for water metering using an ESP-32 microcontroller, a Danfoss MBS 3000 pressure sensor, and a flow meter.
Precisely, they are estimating the water flow by leveraging reasings from pressure sensor.
The prototype is deployed at three locations and data is collected in real-time for further analysis.
They use machine learning to estimate the water flow and helping the stakeholders to take informed decisions and develop startegies for better water management.
With the use of pressure sensors rather than water flow sensors, their prototype is cost-effective.
They have trained three machine learning models -- Linear Regression, Support Vector Regression, and Convolutional Neural Network.

**Strengths:**

*Realworld data collected
*They have trained three ML models on a week's data and show that their models are accurately capturing the groundtruth.

**Weaknesses:**

*It is unclear how authors meaure the groundtruth.
*There should be a discussion on what factors cause non-linear relationships and how to capture them in the machine learning models.
*In its current shape, realtime operation of the system is not possible

---

### Official Review · Reviewer_pJKU · 2023-11-20

**Rating:** 3
**Confidence:** 3

**Summary:**

This study presented an IoT system that utilised machine learning (ML) techniques to estimate water flow through pipes based on pressure. It incorporated an ESP-32 mi- crocontroller, a Danfoss MBS 3000 pressure sensor, and a flow meter deployed at three locations to collect data for three months. ML algorithms modelled the relationship between pressure and flow rate.  The paper is written well and the contribution by comparing different ML techniques is interesting. The paper should be extended by a extended discussion that compares the results and outcomes to the goals outlines in the introduction.

**Strengths:**

This a good subjects. The paper reads well. The introduction is clearly written with references and contributions outlined. The goals are well articulated. A good description of the system and deployment. Data handling is well managed and is the core of the research. The Machine Learning section is interesting, and it is good to compare the performance of different techniques. The discussion and concluding remarks are quite short. The video is informative although quite boring listening the researcher read out the work. It is good to see the data is available via a repository.

**Weaknesses:**

There is other work on placing sensors inside pipes to measure water flow based on pressure and a deeper literature review could have generated a  clearer research question. There is good work achieved in the methodology and through application of ML techniques but unfortunately little evaluation of the results and ML comparison. This paper requires a deeper discussion that revisits each of the goals outlined in the introduction section, currently it is unbalanced.